# Sparse Interaction Additive Networks via Feature Interaction Detection and Sparse Selection

**James Enouen**
Department of Computer Science
University of Southern California
Los Angeles, CA
enouen@usc.edu

**Yan Liu**
Department of Computer Science
University of Southern California
Los Angeles, CA
yanliu.cs@usc.edu

## Abstract

There is currently a large gap in performance between the statistically rigorous methods like linear regression or additive splines and the powerful deep methods using neural networks. Previous works attempting to close this gap have failed to fully investigate the exponentially growing number of feature combinations which deep networks consider automatically during training. In this work, we develop a tractable selection algorithm to efficiently identify the necessary feature combinations by leveraging techniques in feature interaction detection. Our proposed Sparse Interaction Additive Networks (SIAN) construct a bridge from these simple and interpretable models to fully connected neural networks. SIAN achieves competitive performance against state-of-the-art methods across multiple large-scale tabular datasets and consistently finds an optimal tradeoff between the modeling capacity of neural networks and the generalizability of simpler methods.

## 1  Introduction

Over the past decade, deep learning has achieved significant success in providing solutions to challenging AI problems like computer vision, language processing, and game playing [17, 44, 37]. As deep-learning-based AI models increasingly serve as important solutions to applications in our daily lives, we are confronted with some of their major disadvantages. First, current limitations in our theoretical understanding of deep learning makes fitting robust neural networks a balancing act between accurate training fit and low generalization gap. Questions which ask how the test performance will vary given the choice of architecture, dataset, and training algorithm remain poorly understood and difficult to answer. Second, infamously known as blackbox models, deep neural networks greatly lack in interpretability. This continues to lead to a variety of downstream consequences in applications: unexplainable decisions for stakeholders, the inability to distinguish causation from correlation, and a misunderstood sensitivity to adversarial examples.

In contrast, simpler machine learning models such as linear regression, splines, and the generalized additive model (GAM) [16] naturally win in interpretability and robustness. Their smaller number of parameters often have clear and meaningful interpretations and these methods rarely succumb to overfitting the training data. The main shortcoming of these simpler methods is their inability to accurately fit more complex data distributions.

There is a growing strand of literature attempting to merge the interpretability of additive models with the streamlined differentiable power of deep neural networks (DNNs). Key works in this direction such as NAM and NODE-GAM have been able to successfully model one-dimensional main effects and two-dimensional interaction effects using differentiable training procedures [2, 8]. Although many other works have found similar success in modeling one- and two-dimensional interactions, few have made practical attempts towards feature interactions of size three or greater, which we will

36th Conference on Neural Information Processing Systems (NeurIPS 2022).

refer to throughout as higher-order feature interactions. In this way, no existing works have been able to use the hallmark ability of neural networks to model higher-order interactions: amassing the influence of hundreds of pixels in computer vision and combining specific words from multiple paragraphs in language processing. In this work, we bring interpretable models one step closer towards the impressive differentiable power of neural networks by developing a simple but effective selection algorithm and an efficient implementation to be able to train neural additive models which fit higher-order interactions of degrees three and greater. The proposed Sparse Interaction Additive Networks (SIANs) consistently achieve results that are competitive with state-of-the-art methods across multiple datasets. We summarize our contributions as follows:

- We develop a feature interaction selection algorithm which efficiently selects from the exponential number of higher-order feature interactions by leveraging heredity and interaction detection. This allows us to construct higher-order neural additive models for medium-scale datasets unlike previous works in neural additive modeling which only consider univariate and bivariate functions.

- We provide further insights into the tradeoffs faced by neural networks between better generalization and better functional capacity. By tuning the hyperparameters of our SIAN model, we can gradually interpolate from one-dimensional additive models to full-complexity neural networks. We observe fine-grained details about how the generalization gap increases as we add capacity to our neural network model.

- We design a block sparse implementation of neural additive models which is able to greatly improve the training speed and memory efficiency of neural-based additive models. These improvements over previous NAM implementations allow shape functions to be computed in parallel, allowing us to train larger and more complex additive models. We provide code for our implementation and hope the tools provided can be useful throughout the additive modeling community for faster training of neural additive models. [1]

## 2  Related Work

The generalized additive model (GAM) [16] has existed for decades as a more expressive alternative to linear regression, replacing each linear coefficient with a nonparametric function. Two-dimensional extensions appeared in the literature shortly after its introduction [46]. Over the years, an abundance of works have used the GAM model as an interpretable method for making predictions, with the choice of functional model typically reflecting the most popular method during the time period: regression splines, random forests, boosting machines, kernel methods, and most recently neural networks [18, 7, 21, 49]. Two of the most prominent neural network based approaches are NAM and NODE-GA$^2$M [2, 8]. The former stacks multilayer perceptrons to build a one-dimensional GAM; the latter connects differentiable decision trees to build a two-dimensional GAM. Both have demonstrated competitive performance and interpretable trends learned over multiple tabular datasets. Other neural network extensions [47, 33] increase the modeling capacity to higher-order interactions by first training a two-dimensional model and then training an additional blackbox neural network to fit the residual error. While this approach does have higher modeling capacity, it foregoes interpretable insights on the higher-order feature interactions and suffers the same inclination to overfit held naturally by deep neural networks.

Other works in additive modeling instead focus on extending the univariate GAM to sparse additive models in the high-dimensional inference regime [24, 36, 29, 48]. Further extensions of these methods to sparse bivariate models for high-dimensional inference also exist [43, 25]. These works extend classical high-dimensional inference techniques like LASSO and LARS from linear parameters to additive functions by shrinking the effect of minimally important variables and emphasizing sparse solutions. We note that, unlike these works, we do not use sparsity in the soft-constrained sense to shrink features from a fixed selection, but instead adaptively use feature interaction measurements to hierarchically build a feasible set of interactions. The only existing work in additive modeling which uses hierarchy to induce sparsity in the same sense as this work is the GAMI-Net which uses a three stage procedure to select candidate pairs under a hierarchical constraint [49]. Extending their procedure to three or higher dimensions would require four or more stages of training and is left unexplored in their work.

---

[1] Available at github.com/EnouenJ/sparse-interaction-additive-networks

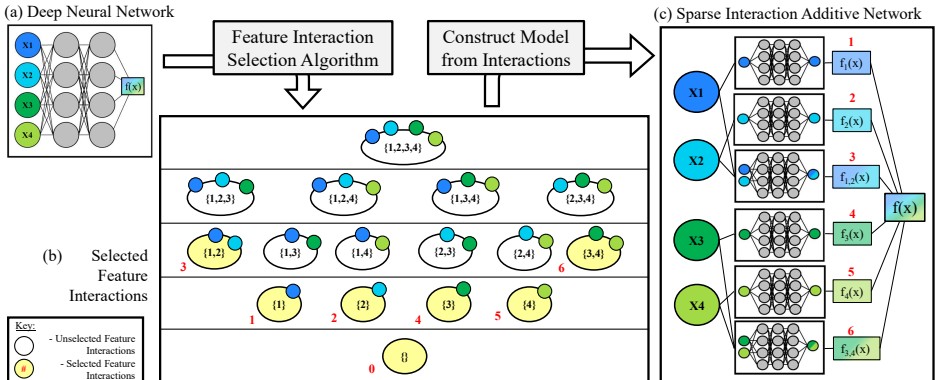

Figure 1: SIAN Pipeline Diagram. (a) We train a base DNN to be able to learn the feature interactions from the dataset given the $d = 4$ input features of $X_1, X_2, X_3, X_4$. (b) We feed our DNN into the FIS algorithm to be able to select from the $2^d = 16$ possible subsets of $\{X_1, X_2, X_3, X_4\}$. (c) We use our selected feature subsets as the GAM array (of length 6 in the image) for our full SIAN network. The empty set function $f_{\{\}}$ is a constant we absorb into the last additive layer of $f(x)$. Finally, we train our SIAN neural network using the specified architecture.

Although the theoretical extension to three-dimensional additive models is clear, there is currently a lack of discussion surrounding the practical challenges faced when trying to model three-dimensional shape functions. One of the few works to pursue practical implementation of higher-order GAMs on real-world datasets is the work of SALSA [21]. This work uses a specialized kernel function to fit additive models of order three and higher. Their work also corroborates our finding that optimal performance is achieved by different orders for different datasets. However, the kernel-based approaches used in this work make it unsuitable beyond small-scale datasets with few samples. This makes our work one of the first to train additive models of higher-order interactions which leverage the automatic differentiation and GPU optimization toolkits which have become commonplace in modern workstations.

## 3 Methods

**Notation** We denote a $d$-dimensional input as $x \in \mathbb{R}^d$ with its $i$-th component as $x_i$; its corresponding output is denoted $y \in \mathbb{R}$. We consider one-dimensional $y$ as in regression and binary classification. We will use $f(x)$ to denote the function or model used to approximate $y$, implicitly considering the additive noise model $y = f(x) + \varepsilon$ for some noise term $\varepsilon$. We denote a subset of the set of features by $\mathcal{I} = \{i_1, \ldots, i_{|\mathcal{I}|}\} \subseteq [d] := \{1, \ldots, d\}$. Its cardinality is denoted $|\mathcal{I}|$, its complement $\setminus \mathcal{I}$, and its power set $\mathcal{P}(\mathcal{I})$. For $x \in \mathbb{R}^d$, we define $x_\mathcal{I} \in \mathbb{R}^d$ such that:

$$(x_\mathcal{I})_i = \begin{cases} x_i & \text{if} \quad i \in \mathcal{I} \\ 0 & \text{otherwise} \end{cases}$$

### 3.1 Generalized Additive Models

We first consider the generalized additive model (GAM) [16], which extends linear regression by allowing each input feature to have a nonlinear relationship with the output.

$$g(y) = f_\emptyset + f_1(x_1) + \cdots + f_d(x_d) \tag{1}$$

Each of the $f_i$ 'reshapes' their respective feature $x_i$ and then adds the reshaped feature to the total prediction. These $f_i$ are hence called shape functions and were traditionally fit using regression splines. The function $g$ is the link function which will be the identity function for regression and the inverse-sigmoid $g(y) = \log(\frac{y}{1-y})$ for classification. $f_\emptyset$ is a normalizing constant. This original formulation where each shape function considers only one feature we will further refer to as GAM-1.

**Feature Interactions**    In order to extend this definition we must consider the interplay which occurs between multiple input features. A 'non-additive feature interaction' between features $\mathcal{I} \subseteq [d]$ for the function $f$ is said to exist when the function $f$ cannot be decomposed into a sum of $|\mathcal{I}|$ arbitrary subfunctions such that each subunction $f_i$ excludes one of the interacting variables $x_i$: $f(x) \neq \sum_{i \in \mathcal{I}} f_i(x_{\{1,2,...,d\} \setminus i})$ [15, 38, 41]. In other words, the entire feature set $\{x_i : i \in \mathcal{I}\}$ must be simultaneously known to be able to correctly predict the output $f(x)$.

The goal of *feature interaction detection* is to uncover these groups of features which depend on one another. For smooth functions, this can be quantitatively done by finding the sets $\mathcal{I} \subseteq [d]$ such that the *interaction strength*, $\omega(\mathcal{I})$, is positive and large.

$$\omega(\mathcal{I}) := \mathbb{E}_x \left[ \frac{\partial^{|\mathcal{I}|} f(x)}{\partial x_{i_1} \partial x_{i_2} \ldots \partial x_{i_{|\mathcal{I}|}}} \right]^2 > 0. \tag{2}$$

We may now adjust the GAM definition to capture feature interactions by considering a set of $T$ specified interactions, $\{\mathcal{I}_t\}_{t=1}^T$, where each $\mathcal{I}_t \subseteq [d]$ is an interaction of size $|\mathcal{I}_t|$:

$$g(y) = f_\emptyset + \sum_i f_i(x_i) + \sum_t f_{\mathcal{I}_t}(x_{\mathcal{I}_t}) \tag{3}$$

The third term extends GAMs to full capacity models which can represent nonlinear dependencies of arbitrary feature sets. For instance, if our set of interactions $\{\mathcal{I}_t\}_{t=1}^T$ includes the complete feature set $[d]$, then our model has exactly the same capacity as the underlying functional model we choose for the shape functions (splines, random forests, deep neural networks, etc.) An abundance of previous works have used this extended version of the GAM model, often called the Smoothing Spline ANOVA model or the functional ANOVA model [7, 36, 29, 18, 46].

Throughout this work, we will use GAM-$K$ to refer to a GAM whose highest order interaction in $\{\mathcal{I}_t\}_{t=1}^T$ is of cardinality $K$. (i.e. $|\mathcal{I}_t| \leq K \ \forall t \in [T]$.) For instance, we will call the NODE-GA$^2$M model [8] a GAM-2 model since the interaction sets are all possible feature pairs: $\{\{i, j\} : i < j \text{ for } i, j \in [d]\}$. We will refer to our SIAN networks as SIAN-$K$ using the same convention.

## 3.2    Feature Interaction Selection

A key concern of using neural networks to fit the shape functions is keeping the number of networks low enough that our training time computation is kept reasonable. While this is typically not a problem for the GAM-1, this can quickly become an issue for the GAM-2. For instance, if we have an input variable $x$ with 30 features, then including all pairwise functions would need to cover the $\binom{30}{2} = 435$ possible pairs. Although learning 465 linear coefficients is reasonable, training hundreds of neural networks becomes less so. Moreover, this quantity only grows exponentially as we increase $K$ to higher-order interactions (3, 4, 5, etc.) In an effort to combat this growing complexity, we introduce a Feature Interaction Selection (FIS) algorithm which depends on two key ingredients: an interaction detection procedure and a heredity condition.

**Feature Interaction Detection**    In recent years, there has been a growing body of work focused on detecting and measuring the strength of feature interactions from large-scale data. Three of the most popular and generally applicable of these methods are the Shapley Additive Explanations (SHAP) [28, 26, 13], Integrated Hessians [40, 19], and Archipelago [42]. For our experiments, we primarily use an adaptation of Archipelago for higher-order interactions because of its compatibility with blackbox models. In contrast, Integrated Hessians is only applicable to sufficiently smooth networks (ReLU networks are not compatible) and SHAP only has fast implementations available for tree-based approaches. Moreover, although both SHAP and Integrated Hessians have clear ideological extensions to higher-order interactions, there are no currently available implementations. A detailed discussion surrounding Archipelago and other detection techniques can be found in Appendix C.

**Heredity**    The practice of only modeling the pairwise interaction effect $\{i, j\} \subseteq [d]$ for some features $i, j \in [d]$ when both of the main effects $\{i\}$ and $\{j\}$ are already being modeled has a long history throughout statistics [34, 11, 5]. There are two main versions of this hierarchical principle explored in the literature: strong heredity and weak heredity. If we are given that $\omega(\{i, j\}) > 0$, strong heredity implies that both $\omega(\{i\}) > 0$ and $\omega(\{j\}) > 0$ whereas weak heredity implies that

$\omega(\{i\}) > 0$ or $\omega(\{j\}) > 0$. For our definition of a feature interaction, we have that strong heredity holds; however, our algorithm will instead focus on a computational version of heredity which asks that $\tau$ percent of subsets are above threshold $\theta$ in order for a pair (triple, tuple, etc.) to be considered as a possible interaction.

**Algorithm**  In Algorithm 1, we detail how we use Archipelago to build our FIS algorithm. The visual overview of the SIAN pipeline is also depicted in Figure 1 above. We start by training a reference DNN which is required for the inductive insights generated by Archipelago. We then pass the trained function approximator to the FIS algorithm along with hyperparameters $K$ (interpretability index), $\tau$ (heredity strength threshold), and $\theta$ (interaction strength threshold). This procedure efficiently searches and ranks the $2^d$ possible feature subsets to produce a final set of interactions which we use to specify the SIAN model architecture. Finally, we train our SIAN neural network using typical gradient descent methods. In Appendix A, we provide a further theoretical discussion in which we prove exact recovery of the true feature interactions and show how this sparse selection leads to provably lower generalization gaps in a toy additive noise model.

### 3.3   Block Sparse Implementation

In order to improve the time and memory efficiency of the SIAN model, we implement a block sparse construction for neural additive models. The default scheme of SIAN is to use a network module for each of the shape functions and additively combine the output features, following the implementation strategies of NAM and other popular neural additive models. However, since each shape function network is computed sequentially, this greatly bottlenecks the computation speed of the model. We instead construct a large, block sparse network which

---

**Algorithm 1** Feature Interaction Selection (FIS)

**Inputs**: Trained prediction model $f(x)$, and validation dataset $X^* = \{x^{(1)}, \ldots, x^{(V)}\}$
**Parameters**: Cutoff index $K$, cutoff threshold $\tau$, strength threshold $\theta$
**Output**: $\mathcal{I}$, a family of feature interactions with index at most $K$ and strength above $\theta$

1:  Set $\mathcal{I} \leftarrow \emptyset$
2:  Set $\mathcal{J} \leftarrow \{\{i\} : i \in [d]\}$
3:  $k \leftarrow 1$
4:  **while** $k \leq K$ **do**
5:      **for** $J$ in $\mathcal{J}$ **do**
6:          $\omega_J(X^*) \leftarrow 0$
7:          **for** $x^{(v)} \in X^*$ **do**
8:              $\omega_J(x^{(v)}) \leftarrow \text{Archipelago}(f, x^{(v)}, J)$
9:              $\omega_J(X^*) \leftarrow \omega_J(X^*) + \omega_J(x^{(v)})$
10:         **end for**
11:         $\omega_J(X^*) \leftarrow \omega_J(X^*) \cdot \frac{1}{|X^*|}$
12:         **if** $\omega_J(X^*) > \theta$ **then**
13:             $\mathcal{I} \leftarrow \mathcal{I} \cup \{J\}$
14:         **end if**
15:     **end for**
16:     $\mathcal{J}' \leftarrow \{J : J \in \mathcal{P}([d]); |J| = k + 1\}$
17:     **for** $J$ in $\mathcal{J}'$ **do**
18:         $\nu(J) \leftarrow \frac{1}{|J|} \sum_{I \subseteq J} [1_{I \in \mathcal{I}} \cdot 1_{\{|J| - |I| = 1\}}]$
19:     **end for**
20:     $\mathcal{J} \leftarrow \{J : J \in \mathcal{J}'; \nu(J) > \tau\}$
21:     $k \leftarrow k + 1$
22: **end while**
23: **return** $\mathcal{I}$

---

computes the hidden representations of all shape function networks simultaneously, leveraging the fact that each shape subnetwork has the same depth. Using this block sparse network allows for shape features to be computed in parallel, leading to a significant improvement in training speed. The main consequence of this design is a higher footprint in memory; therefore, we also develop a compressed sparse matrix implementation of the network which has a greatly reduced memory footprint for saving network parameters. SIAN is able to interchange between these different modes with minimal overhead, allowing for faster training in the block sparse module, lower memory footprint in the compressed module, and convenient visualization of shape functions in the default module. We provide further numerical details of our gains in Section 5.1.

## 4   Datasets

Our experiments focus on seven machine learning datasets. Two are in the classification setting, the MIMIC-III Healthcare and the Higgs datasets [20, 3]. The other five are in the regression setting, namely the Appliances Energy, Bike Sharing, California Housing Prices, Wine Quality, and Song Year datasets [6, 14, 22, 12, 4]. More details about each dataset are provided in Table 1. We evaluate the regression datasets using mean-squared error (MSE). We measure the performance on the classification datasets using both the area under the receiver operating characteristic (AUROC) and

the area under the precision-recall curve (AUPRC) metrics. We report both metrics for the MIMIC dataset since the positive class is only 9% of examples and report only AUROC for the Higgs dataset since it is relatively well-balanced.

## 4.1 Experiment Details

For the baseline DNNs we are using hidden layer sizes [256,128,64] with ReLU activations. For the GAM subnetworks we are using hidden layer sizes [16,12,8] with ReLU activations. We use L1 regularization of size 5e–5. In the main results section, we report the results for each $K \in \{1, 2, 3, 5\}$ using only a single value of $\tau$ and $\theta$. The hyperparameter $\tau$ was taken to be $0.5$ throughout and $\theta$ was selected from a handful of potential values using a validation set. We train all networks using Adagrad with a learning rate of 5e–3. All models are evaluated on a held-out test dataset over five folds of training-validation split unless three folds are specified. Three folds are used for NODE-GAM on all datasets as well as Song Year and Higgs for all models. We respect previous testing splits when applicable, otherwise we subdivide the data using an 80-20 split to generate a testing set. In addition to NODE-GA2M, we compare against the interpretable models LASSO and GA$^2$M EBM as well as the popular machine learning models of support vector machines (SVM), random forests (RF), and extreme gradient boosting (XGB) [10, 30].

Table 1: Real-world datasets. $n$ is the number of samples, $d$ is the number of features, and $p$ is the percentage of data samples with the positive class label.

| Dataset | $n$ | $d$ | $p$ |
|---|---|---|---|
| Higgs Boson | $11,000,000$ | $28$ | $53.0\%$ |
| MIMIC-III | $32,254$ | $30$ | $9.2\%$ |
| Energy Appliances | $19,735$ | $30$ | – |
| Bike Sharing | $17,379$ | $13$ | – |
| California Housing | $20,640$ | $8$ | – |
| Wine Quality | $6,497$ | $12$ | – |
| Song Year | $515,345$ | $90$ | – |

Table 2: MIMIC-III Performance.

| Model | AUROC (⇑) | AUPRC (⇑) |
|---|---|---|
| SAPS II | 0.792 | 0.281 |
| SOFA | 0.703 | 0.225 |
| LASSO | 0.568 | 0.396 |
| GA$^2$M EBM | 0.840 | 0.375 |
| NODE-GA$^2$M | 0.826 | 0.345 |
| SIAN-1 | 0.848 | 0.409 |
| SIAN-2 | **0.855** | **0.423** |
| SIAN-3 | **0.856** | **0.425** |
| SIAN-5 | **0.856** | **0.425** |
| RF | 0.821 | 0.369 |
| SVM | 0.831 | 0.404 |
| XGB | 0.843 | 0.382 |
| DNN | 0.844 | 0.382 |

# 5 Results

Across seven different datasets, SIAN achieves an average rank of 3.00 out of the 8 models we consider. The next best performing model, NODE-GA2M, has an average rank of 3.71 out of 8. The third best performing model, DNN, has an average rank of 3.86 out of 8. We find that SIAN achieves consistent performance by being able to adapt to both the low-dimensional datasets and the high-dimensional datasets, finding a balance between good training fit and good generalization.

Table 3: Test metrics for six of seven datasets. (⇑)/(⇓) indicates higher/lower is better, respectively.

| Model | Appliances Energy (⇓) | Bike Sharing (⇓) | California Housing (⇓) | Wine Quality (⇓) | Song Year (⇓) | Higgs Boson (⇑) |
|---|---|---|---|---|---|---|
| LASSO | 0.740±0.002 | 1.053±0.001 | 0.478±0.000 | 0.575±0.002 | 1.000±0.008 | 0.635±0.000 |
| GA$^2$M EBM | 1.053±0.138 | 0.124±0.004 | 0.265±0.002 | 0.498±0.004 | 0.894±0.001 | 0.698±0.001 |
| NODE-GA$^2$M | 1.064±0.056 | **0.111±0.006** | **0.222±0.005** | 0.521±0.009 | 0.806±0.001 | 0.811±0.000 |
| SIAN-1 | **0.718±0.007** | 0.387±0.035 | 0.378±0.007 | 0.551±0.004 | 0.860±0.001 | 0.771±0.001 |
| SIAN-2 | 0.763±0.009 | 0.127±0.008 | 0.302±0.002 | 0.497±0.003 | 0.842±0.002 | 0.795±0.001 |
| SIAN-3 | 0.808±0.026 | 0.125±0.013 | 0.278±0.001 | 0.497±0.003 | 0.831±0.001 | 0.798±0.001 |
| SIAN-5 | 0.801±0.031 | 0.149±0.011 | 0.272±0.003 | 0.484±0.006 | 0.821±0.001 | 0.802±0.001 |
| RF | 1.114±0.095 | 0.206±0.009 | 0.271±0.001 | **0.439±0.005** | 0.994±0.005 | 0.654±0.002 |
| SVM | 0.740±0.008 | 0.168±0.001 | 0.262±0.001 | 0.457±0.008 | 0.940±0.012 | 0.698±0.001 |
| XGB | 1.188±0.119 | 0.157±0.003 | 0.229±0.002 | 0.465±0.014 | 0.881±0.002 | 0.740±0.000 |
| DNN | 0.945±0.054 | 0.374±0.017 | 0.283±0.005 | 0.495±0.007 | **0.791±0.002** | **0.823±0.000** |

For the MIMIC dataset, we can see the models' performances in terms of both AUROC and AUPRC in Table 2. In addition to the models we previously described, we also compare against the interpretable medical baselines of SOFA and SAPS II which are simple logic-based scoring methods [23, 45]. We see that the machine learning methods improve over the baseline performance achieved by the SAPS and SOFA methods.

In Table 3, we can see the combined results for our six other datasets. First, in the Appliances Energy dataset, we see that the SIAN-1 performs the best, with LASSO and SVM trailing slightly behind. The success of one-dimensional methods on this dataset could be indicative that many of the dataset's trends are one-dimensional. As we increase the dimension of the SIAN, the test error slowly increases as the generalization gap grows; the full-complexity DNN has even worse error than all SIAN models.

Second, in the Bike Sharing dataset, we see that the best performing model is the NODE-GA2M, with the EBM, SIAN-2, and SIAN-3 trailing only slightly behind. All of these methods are two or three dimensional GAM models, again hinting that a significant portion of the important trends in this dataset could be bivariate. Indeed, the most important bivariate trend accounts for more than $50\%$ of the variance in the dataset, as we explore in Figure 2 below.

For the remaining datasets, we see that the performance of the SIAN model improves as we add higher and higher-order feature interactions. For the California Housing dataset, we find the best performance using the differentiable tree method of NODE-GA2M. For the Wine Quality dataset, we find the best performance using the random forest algorithm. For the larger-scale datasets of Song Year and Higgs Boson, we find that the best performance is still obtained by a full-complexity deep neural network. These two are the only datasets where SIANs of order five or less are not sufficient to outperform vanilla deep neural networks, implying there are important feature interactions of degree greater than five.

## 5.1 Training Speed and Storage

In Table 4 below we see how our SIAN network compares against other popular differentiable additive models in both training time and size on disk. For fair comparison we do not utilize our interaction selection algorithm for SIAN in this section, instead training SIAN-2 with all possible pairs. We see that our implementation of different modes for the SIAN architecture allows us to outperform both NAM and NODE-GAM, with 2-8x faster training on GAM-1 models and 10-80x faster training on GAM-2 models. We reiterate that because the overhead for switching between modes is negligible, SIAN enjoys the benefits of all modes: training quickly in the block sparse mode and saving succinctly in the compressed mode.

Table 4: Training Time and Memory Size. Experiments are run with a machine using a GTX 1080 GPU and 16GB of RAM. The average wall-clock time over three runs is reported.

| | | SIAN-2 (default) | (block-sparse) | (comp-ressed) | NODE-GA2M | SIAN-1 (default) | (block-sparse) | (comp-ressed) | NAM |
|---|---|---|---|---|---|---|---|---|---|
| Training Time (minutes) | Wine | 17.57 | **0.69** | – | 55.19 | 3.03 | **0.66** | – | 5.45 |
| | Bike | 159.65 | **5.38** | – | 60.75 | 25.65 | **4.58** | – | 12.09 |
| | House | 88.67 | **6.09** | – | 58.03 | 22.72 | **5.88** | – | 14.74 |
| Size on Disk (KB) | Wine | 7,113 | 7,113 | **537** | 1,040 | 182 | 182 | **86** | 1,203 |
| | Bike | 9,727 | 9,666 | **626** | 1,040 | 218 | 212 | **92** | 308 |
| | House | 1,526 | 1,526 | **251** | 1,040 | 83 | 83 | **59** | 1,921 |

## 5.2 Beyond ReLU Networks

The FIS algorithm we describe can be applied to other combinations of FID algorithm + functional model besides Archipelago + ReLU Neural Networks. To demonstrate the general applicability of our scheme, we replace the continuous, piecewise-linear functions of ReLU neural networks with the piecewise-constant, differentiable decision trees using NODE-GAM. We extend the original implementation of NODE-GAM to handle feature triplets, extending the method to a trivariate function or GAM-3 model. We run our FIS algorithm using $K = 3$ on both the Housing and Wine datasets to fit GAM-3 models using the inductive biases of the NODE architecture.

Extending from NODE-GA2M to NODE-GA3M is able to improve performance from $0.222$ to $0.175$ on the Housing dataset, a $21.2\%$ further improvement over the state of the art method. The same extension is only able to deliver a $4.8\%$ improvement from $0.521$ to $0.496$ on the Wine dataset; however, both the SIAN-5 and NODE-GA3M are able to improve performance over all previously available additive models. These two examples demonstrate the ability of our FIS pipeline to be applied to more general machine learning techniques to model higher-order interactions.

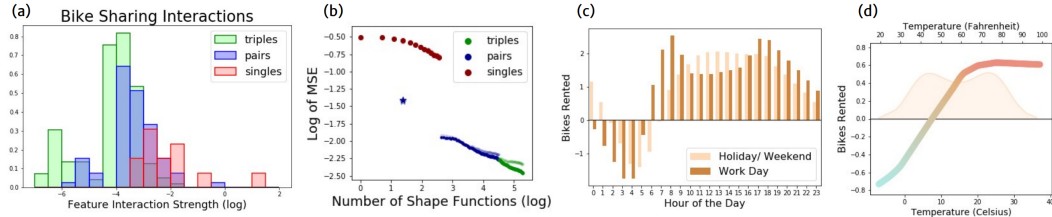

Figure 2: Analysis of Bike Sharing. (a) We plot the histogram of feature interaction strength, adjusting the vertical scaling by degree to display in a single plot. (b) We plot the training and validation performance (darker and lighter, respectively) as we add feature interactions to SIAN. We use an exponentially weighted moving average to reduce variance instead of training multiple networks of each size. The blue star is the performance of the shape functions in 2c and 2d. (c) The most important shape function showing the interaction between hour of the day and work day. (d) The next shape function showing temperature's effect on bikes rented.

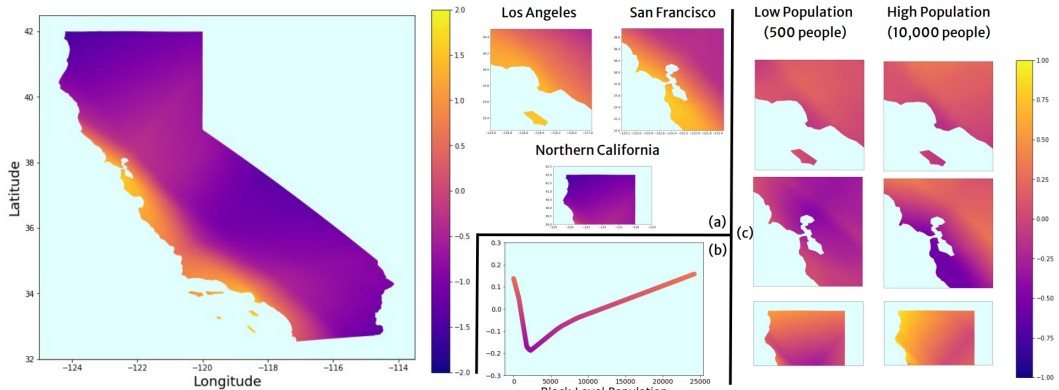

Figure 3: 3D feature interaction for California Housing. (a) The effect of the location only for the entire state of California and zoomed in to three selected locations. (b) The effect of only the block-level population. (c) The three-dimensional interaction effect of block-level population with location. We plot the effect for both a low and high population to highlight the differences in effect for each location. Note the different scales per panel.

## 6 Discussion

In this section, we further explore the feature interactions learned by SIAN and FIS across multiple datasets. We provide multiple visualizations of the shape functions learned by SIAN to get a sense of the diverse analysis and interpretations which are made possible by additive models. Further discussion and graphics are provided in Appendix C and D.

In Figure 2a, we visualize the interaction strength measured for each of the singles, pairs, and triples of features from the Bike Sharing dataset. The four top rated interactions (three of which are visually separated from the main body of the histogram) are, from right to left, ["hour", "hour X workday", "workday", "temperature"]. In 2b we see the effect of gradually adding feature interactions to our model. We see there is a steep jump in performance when we are able to model the first pair, depicted in 2c. Together with 2d, these two visualizations alone explain 84% of the variance in the Bike Sharing dataset. Importantly, these two trends are interpretable and agree with our intuition about when people are more likely to bike: on work days, we see peak spikes at 8 a.m. and 5 p.m., corresponding to the beginning and end of work hours; on weekends and holidays, there is a steady demand of bikes throughout the afternoon. There are also more bikers during warmer temperatures.

In Figure 3, we set out to visualize one of the three dimensional interactions which occurs between the three features of latitude, longitude, and population. In 3a, we see the rise in housing price along the coast of California, especially around the metropolitan areas of Los Angeles and San Francisco. In 3b, we see that house price does not monotonically increase with population as we might expect for higher population densities. A detailed look into the dataset reveals that the 'population' feature being used is accumulated at the census block level, creating an inverse relationship with population density as areas like LA and SF are subdivided more than their suburban and rural counterparts. Although it is

possible location and population density might have independent effects on housing price, the dataset nevertheless induces an interaction between location and population. In 3c, we attempt to visualize this 3D interaction by viewing subsamples across two population sizes and over three regions. We see the network has learned to differentiate the urban regions of LA and SF from the rural coast of northern California, where high population becomes indicative of higher population densities and higher housing prices. We note that the trends learned by SIAN tends to be very continuous, which is a potential shortcoming in learning fine-grained block-level information in cities like Los Angeles and San Francisco. In light of these concerns, we run experiments on a three-dimensional extension of the piecewise constant, discontinuous NODE-GAM model in section 5.2.

In Figure 4, we see two example trends from the SIAN trained to predict mortality given hospital data. On the left, we see that health risk gradually increases with age. On the right, we see the trend with respect to the Glasgow Coma Scale indicator which is a measure of alertness and consciousness. The lowest and highest scores correspond to severe head injury and full alertness, respectively.

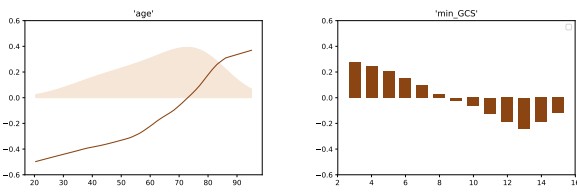

Figure 4: MIMIC-III. Age Risk and GCS Risk

We see that very low GCS corresponds to high risk and that the risk decreases as GCS increases. At the score of 13, however, we see that an increase to 14 or 15 actually increases mortality risk, defying the intuition that risk should be monotone in the GCS score. It is highly likely that this dip in mortality risk comes from the special care given to patients with GCS scores below 15, compared to their counterparts with perfect scores.

This trend illustrates a phenomenon occurring throughout machine learning applications in healthcare where correlation and causation are conflated with one another. Similar results have been found linking asthma to a decreased risk of death from pneumonia [7]. These issues might go unnoticed in black-box models whereas interpretable models can uncover and resolve such discrepancies before deployment. While discovering and correcting interpretable trends brings clear advantages, it is also possible for these trends to be misinterpreted by non-experts as a causal relationship. Such false causal discoveries can not only lead to physical harm in the domain of medicine, but also to larger social harm in broader AI systems and applications.

## 7 Limitations and Future Work

A primary limitation of the current work is its focus on multilayer perceptrons whereas modern state-of-the-art results are dominated by geometric deep learning and transformer architectures. Extending this procedure to more general architectures is a key direction for bringing interpretability to domains like computer vision and natural language. Such domains can further customize the FIS algorithm to respect specific structures like spatial locality and knowledge graph semantics.

Another important direction of research includes a better theoretical understanding how the benefits of SIAN scale with the dimensionality of the dataset and the number of samples provided, providing yet another lens to study the implicit biases of deep neural networks. Further, developing a theory which accurately models the empirically observed distributions of feature interactions in real-world data, especially in the presence of heteroscedastic noise and correlated features, would be of great interest for this direction.

Multiple experiments confirm that SIAN can produce powerful and interpretable machine learning models which match the performance of state-of-the-art benchmarks like deep neural networks and NODE-GA2M. Further experiments show that FIS can be applied to more general machine learning algorithms. Hopefully, future work will be able to further clarify this sparse interaction perspective and help deepen our understanding of the generalization performance of neural networks.

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
