# OpenReview forum: "Sparse Interaction Additive Networks via Feature Interaction Detection and Sparse Selection"
_NeurIPS.cc/2022/Conference — NeurIPS 2022 Accept_

### Official Review · Reviewer_wCbJ · 2022-07-11

**Rating:** 6
**Confidence:** 3
**Soundness:** 4 excellent
**Presentation:** 4 excellent
**Contribution:** 3 good

**Summary:**

The paper discusses how to generalize neural additive models to higher-order interactions and how to do them efficiently. This is done in three steps: 1) a general DNN is trained; 2) a feature interaction algorithm based on higher-order derivatives with a large norm is used to choose which features to focus on for learning neural additive models. These are then applied to an additive network to learn the final features. Experimental results demonstrate reasonable performance on regression and classification datasets

**Questions:**

Q1: Table 2 is a bit confusing. Both regression and classification metrics and datasets are listed, and for one higher is better, for the other lower is better. It would be clearer to clearly indicate which is which, as the reader might have a hard time switching context between Section 4 and it. Also, it is not all datasets since MIMIC III is in Table 3.

Q2: It might be nice to show the visualization in the discussion via some "automated" process, as it would illustrate that these type of data analysis/visualization can be done automatically (what are the shared components in these post-processing steps that can be written into programs easily?). The results in the discussion section are nice, but it is also possible to produce these with some "manual" exploratory data analysis methods.

**Limitations:**

Yes

**Strengths And Weaknesses:**

Strength:
- A sound solution for addressing the efficiency issues in selecting higher-order interactions for generalized additive models.
- Extensive experiments on various classification and regression datasets.
- The explanation of MIMIC III is particularly interesting.
- Block sparse implementations massive improves training, and compression reduces memory cost.

Weaknesses:
- Relatively weak direct technical novelty as it mostly combines Archipelago and generalized additive models.

---

> ### Author Response · Authors · 2022-08-02
> **Thank you for your very considerate review and understanding of key strengths**
>
> > Table 2 is a bit confusing…
>
> That is a very good point regarding table 2, we have updated the table with arrows for each dataset to help the reader more easily interpret the results.
>
> > It might be nice to show the visualization in the discussion via some "automated" process
>
> You will be glad to know that our codebase can automatically visualize the learned shape functions from a SIAN model in a style similar to Figure 4. (Other figures were given visual enhancement for the manuscript.)  We have previously considered visualizing the entire SIAN-2 model for MIMIC-III in the appendices and have now added it to a new section Appendix D.
>
> > Relatively weak direct technical novelty as it mostly combines Archipelago and generalized additive models.
>
> Regarding your main concern of technical novelty, we certainly admit that both Archipelago and GAMs are existing methods, however, it has been standard practice for decades to fit GAMs using all available terms.  Historically, this has limited their application to only 1D or 2D functions.  Even as recently as 2021, prominent papers fitting GAMs using DNNs have used the mantra of “train then remove” rather than using a feature interaction selection algorithm before training (see, for example [49] and [50]).  To the best of our knowledge, despite the recent surge in interest in neural additive models, no existing work has been unable to train 3D functions and higher.  It is for this reason we believe there is novelty not in the individual methods, but in our application of the FIS algorithm to GAMs, where we combine Archipelago with heredity in a novel way.
>
> > it is also possible to produce these with some "manual" exploratory data analysis methods.
>
> Yes, it is partially true that manual data exploration can produce plots similar to the ones in the discussion section.  We have added a brief section C.3 to illustrate the differences a little further.  The main difference between SIAN and posthoc attribution of an uninterpretable model is that SIAN gives a global explanation of exactly why the model makes its predictions, whereas attribution only gives a local approximation of the decision, centered around the given data point.  In terms of visualization, this essentially results in a scatter plot of attribution, rather than a heatmap or continuous plot.  Often, these scatter plots are too noisy on their own to see the real trend, and fitting a model to these attributions could be met with the question: why not just fit an additive model in the first place?
>
> [49] "Sparse Neural Additive Model: Interpretable Deep Learning with Feature Selection via Group Sparsity"
> [50] "GAMI-Net: An Explainable Neural Network Based on Generalized Additive Models with Structured Interactions"

---

> > ### Comment · Reviewer_wCbJ · 2022-08-08
> > **Thank you for the response.**
> >
> > Thank you for the response and revision to the paper. I think my questions are properly addressed.

---

> > > ### Author Response · Authors · 2022-08-09
> > > **Thank you for the review**
> > >
> > > Thank you for helping us improve the manuscript with your suggestions.  We hope we were able to sufficiently answer the majority of your questions.

---

### Official Review · Reviewer_FZvZ · 2022-07-11

**Rating:** 6
**Confidence:** 2
**Soundness:** 3 good
**Presentation:** 3 good
**Contribution:** 3 good

**Summary:**

This paper proposed Sparse Interaction Additive Networks (SIAN), which can effectively tackle the exponential number of high-order feature interactions and scale up to larger scale datasets. By leveraging heredity and interaction detection, SIAN achieves competitive performance across multiple datasets. The work also provides further insights into the generalization and capacity trade-off and a block sparse implementation of neural additive models, which achieve better training speed and memory efficiency of neural-based additive models.

**Questions:**

How do the training speed and storage change along with the increasing of feature intersection orders in SIAN ?

**Limitations:**

The authors have adequately addressed the limitations and potential negative societal impact of their work.

**Strengths And Weaknesses:**

#### Strength:

1. The paper is well organized and easy to follow.
2. Sufficient implementation details are provided for reproduction.
3. The experiment section consists of both quantitative results and several visualization analyses.

#### Weakness:

1. As shown in Table 2, SIAN-1 performs the best in Appliances Energy dataset, while SIAN-2/3 is better in Bike Sharing and SIAN-5 in the rest. So how to effectively choose the feature intersection order used in SIAN for better performance.
2. When we increase the intersection order of additive models, will it reduce the model's interpretability to some extent, which may degrade the value of performance improvement.
3. Letter case should remain consistent, e.g., sian-1/2/3/5 in Table2, but SIAN in the main text.

---

> ### Author Response · Authors · 2022-08-02
> **Thank you for tending to accept this paper**
>
> > So how to effectively choose the feature intersection order used in SIAN for better performance.
>
> The feature interaction order for best performance can easily be selected by looking at the validation performance.  We find that as a function of K, the performance is typically a convex function: decreasing as we add training capacity but ultimately increasing as we begin to overfit.  This pattern can be seen in all datasets and visualized at greater resolution in Figure 2b.
>
> > will it reduce the model's interpretability to some extent
>
> As you importantly mention, when we increase the feature interaction order, the shape functions become more difficult to interpret.  Previously, practitioners are forced to choose between the overly simple but interpretable bivariate models or the complex but uninterpretable deep networks.  Our work is able to bridge/ interpolate between these two extremes, allowing practitioners greater control over the interpretability-accuracy tradeoff to suit their target application.
>
> > How do the training speed and storage change along with the increasing of feature intersection orders in SIAN ?
>
> There is a linear increase in storage size as we increase the number of feature interactions, but our block sparse formulation allows for sublinear increases in training time.  For example, on the Wine dataset in Table 4: increasing the interactions from 12 singles to 78 pairs and singles (6.5x increase) results in a size increase of 86KB to 537KB (6.24x increase) whereas the training time only increases from 40 seconds to 42 seconds.  This demonstrates the significant improvement in speed from our block sparse technique.
>
> Nevertheless, when considering higher-order interactions, this might still be insufficient.  For the Song Year dataset, there are as many as 44 million possible quintuples to consider (90 choose 5).  Training even a linear regression model with this many terms becomes infeasible.  It is for this reason we develop the FIS selection algorithm which leverages Archipelago and heredity in a novel way to automatically select the most important feature interactions.  Using this small selection of important feature interactions, we can easily train a SIAN-5 model.

---

> > ### Comment · Reviewer_FZvZ · 2022-08-07
> > **Thanks for the response**
> >
> > I would thank the authors for the detailed responses, which addressed all my concerns. Thus I tend to accept this paper and keep my score.

---

> > > ### Author Response · Authors · 2022-08-09
> > > **Thank you**
> > >
> > > We are happy to have addressed all of your major concerns.

---

### Official Review · Reviewer_Q2gD · 2022-07-12

**Rating:** 6
**Confidence:** 3
**Soundness:** 2 fair
**Presentation:** 3 good
**Contribution:** 2 fair

**Summary:**

This paper proposes Sparse Interaction Additive Networks (SIANs). On an array of tabular datasets, SIANs perform comparably to neural networks and offer more interpretability than neural networks.

**Questions:**

1. In Tables 2, 3, and 4, are DNN baselines here also the reference DNNs used in Feature Interaction Selection (FIS)?

2. Would it be possible to have a discussion / empirical comparison between the interpretability of SIANs and DNNs undergone posthoc feature-attribution? As a concrete example, the authors can train a standard DNN on the California Housing dataset and visualize key features crucial to its performance. Are these features meaningful as well? And do they give rise to the same interpretaion as by SIANs in Figure 3?

**Limitations:**

In my opinion, the interpretability of SIANs comes at the cost of being only able to deal with tabular datasets. In fact, even for tabular datasets, it is unclear to me whether the interpretability of SIANs is indeed greater than equipping conventional DNNs with feature attribution methods -- I encourage the authors to provide such a comparison.

**Strengths And Weaknesses:**

**Strength**. The paper is very well written. The goal of the paper -- scaling up interpretable additive models while making their training to be tractable is very well motivated.

**Weakness**. In my opinion, this paper can be further improved by incorporating a discussion / empirical comparison between the interpretability of DNNs and SIANs. To my understanding, the main advantage of SIANs is not their data-fitting ability (DNNs are better in these perspectives) but their interpretability. However, there has been a thrust of research in the post-hoc interpretation of neural networks such as saliency maps. These DNN-based saliency maps (or other attributed features) give rise to meaningful interpretation while being very flexible: They can be applied to DNNs in a model-agnostic way. The applicability of SIANs, however, is currently confined to tabular datasets and MLPs.

---

> ### Author Response · Authors · 2022-08-02
> **Thank you for your review**
>
> > To my understanding, the main advantage of SIANs is not their data-fitting ability (DNNs are better in these perspectives) but their interpretability.
>
> While we agree that DNNs have a higher functional capacity, we believe that SIAN has not only higher interpretability, but higher robustness.  In Figure 2b, we can see when we increase the functional capacity of SIAN (to be closer and closer to that of a DNN), we decrease the robustness of the SIAN model (note the widening gap between training/validation performance.)  This ultimately leads to SIAN having a higher effective data-fitting ability, because it does not easily overfit to spurious correlations.  Please note that SIAN outperforms MLPs on 5 out of the 7 datasets we consider.
>
> > Would it be possible to have a discussion / empirical comparison between the interpretability of SIANs and DNNs undergone posthoc feature-attribution?
>
> Yes, we have added a brief section to the Appendix C.3 discussing this.  Importantly, the key distinction between SIAN/ additive model interpretability and posthoc feature interpretability is that SIAN is a global method while feature attribution is a local method.  Therefore, while SIAN displays the exact decision-making process of the model, posthoc attribution only gives a local approximation of the feature importance given a specified data sample.  This means that to produce a similar plot to Figure 3, we can only consider a scatter plot of training samples instead of a heatmap of the true model.  Oftentimes, these scatter plots can be too noisy to discern a clear trend as in Figure 3.  Further, these attributions can sometimes be misleading since MLPs do not naturally disentangle their features from one another, see [1], [2].
>
> > The applicability of SIANs, however, is currently confined to tabular datasets and MLPs.
>
> Although it is the primary focus of this work, we feel that the framework of SIAN is ultimately more general than both MLPs and tabular data.  In section 5.2, we apply our framework to a different type of model besides MLP models, using instead the NODE architecture and NODE-GAM setup.  Using this setup is able to advance state-of-the-art on one of the datasets we consider.  We mainly focus on tabular datasets in this work because of their easily interpretable features; however, the techniques from SIAN can likely extend beyond tabular data.  In preliminary experiments on MNIST, we have accuracies of: MLP, 98.9%; CNN, 99.6%; SIAN, 99.3%.
>
>
> Ultimately, a comparison of interpretability is fundamentally both a task-dependent and a subjective matter.  Nevertheless, we hope we have addressed many of your concerns and can follow up with additional details surrounding these questions.
>
> [1] “Sanity Checks for Saliency Maps”
> [2] “Stop Explaining Black Box Machine Learning Models for High Stakes Decisions and Use Interpretable Models Instead”

---

> > ### Comment · Reviewer_Q2gD · 2022-08-06
> > **Thank you for your rebuttal**
> >
> > I'd like to thank the authors for their response. Especially, I find the discussion of the (newly added) Appendix C.3 very helpful. I have raised my score to 6.
> >
> > Sorry if this sounds like a classic reveiwer #2 cliche, but I think that the paper can be further improved if the authors could test SIAN on harder tasks beyond MNIST, e.g., on CIFAR. Apparently, both SIAN and baselines solve the MNIST task near-perfect (~99% accuracy), implying that this benchmark is pretty much saturated. Training on a harder task will either further demonstrate the impressive performance of SIAN or reveal its limitation -- either way, it would be a great contribution to the community.

---

> > > ### Author Response · Authors · 2022-08-09
> > > **Thank you!**
> > >
> > > We greatly appreciate your reconsideration and are glad to have addressed your concerns regarding interpretability. Also, we are further working on applying these techniques to other specific tasks, including computer vision, which were previously unthinkable for additive models. One of the key reasons to keep such a work separate is because of the need to focus on vision-specific interpretability and vision-specific architectures.

---

### Meta-Review · Area_Chair_VnMp · 2022-08-27

**Recommendation:** Accept
**Confidence:** Certain

**Metareview:**

This paper proposes a scheme to augment a trained neural network (considering in particular the case of unstructured, tabular data) by extending generalized additive models to the multi-layer neural setting in an unusual manner by using higher-order derivatives from an initial deep neural network to select a sparse set of higher order feature interactions on which to fit their augmented network.

Reviewers considered the paper well written, easy to follow, considered the method sound and well-motivated, and praised experiments as thorough and detail as adequate for reproducibility. Q2gD wondered specifically how the interpretability of these models measures against post-hoc DNN interpretation methods; the authors responded with a new section in the appendix, causing Q2gD to raise their score. FZvZ had questions about the selection of the model order and how that might affect interpretability which were adequately addressed in rebuttal. wcBj points to "relatively weak direct technical novelty", to which the authors reasonably respond that their forward selection method for interaction terms stands apart from typical approaches that involve backward selection or pruning; several confusing aspects and an suggestion for an explanatory visualization comprises a new section in the Appendix.

The experimental results seem well chosen and are convincing, even if the proposed method SIAN is not uniformly the best method on all tasks considered, it is a strong contender overall and in several cases handily outperforms the DNN baseline. The selection procedure seems well motivated and clever, while the architecture of the resultant additive model seems like one particular choice in a sea of possibilities. I doubt this paper will be the last word on the matter. Nonetheless, this seems like a valuable contribution to the literature on applying DNNs to tabular data and the intersection of GAM techniques with deep learning. I thus recommend acceptance.

**Award:**

No

---

### Decision · Program_Chairs · 2022-09-14

Accept